# A compliant metastructure design with reconfigurability up to six degrees of freedom

Humphrey Yang[1], Dinesh K. Patel [1,2], Tate Johnson [1,3], Ke Zhong[1,4], Gina Olson[5], Carmel Majidi [2], Mohammad F. Islam [4], Teng Zhang [6,7] & Lining Yao [1,8]

Compliant mechanisms with reconfigurable degrees of freedom are gaining attention in the development of kinesthetic haptic devices, robotic systems, and mechanical metamaterials. However, available devices exhibit limited programmability and form-customizability, restricting their versatility. To address this gap, we propose a metastructure concept featuring reconfigurable motional freedom and tunable stiffness, adaptable to various form factors and applications. These devices incorporate passive flexures and actively stiffness-changing rods to modify kinematic freedom. A rational design pipeline informs the flexures' topological arrangements, geometric parameters, and control signals based on targeted mobilities, enabling the creation of unitary joints with up to six degrees of freedom. Our demonstrative application examples include a wrist device that has an effective stiffness of 0.370 Nm/deg (unlocked state, 5% displacement) to 2.278 Nm/deg (locked state, 1% displacement) to enable dynamic joint mobility control, a haptic thimble device (2.27-52.815 Nmm$^{-1}$ at 1% displacement) that mimics the sensation of touching physical materials ranging from soft gel to metal surfaces, and a wearable device composed of multiple joints tailored for the arm and hand to augment haptic experiences or facilitate muscle training. We believe the presented method can help democratize compliant metastructures development and expand their versatility for broader contexts.

Mechanical systems that afford tunable kinematics and stiffness have been envisioned to augment virtual haptic interactions[1–4], productivity[5–9], and medical rehabilitation or assistance[10–14]. Along this line of research, literature has explored engineering methods for creating stiffness reconfigurable mechanisms or metamaterials[15–18], as well as methods for designing stiffness reconfiguration along multiple targeted degrees of freedom (DOF)[19–28]. While such kinematic and stiffness reconfigurability could often be achieved through electromagnetic[15,19,24,25,29–32], electrostatic[33,34], or pneumatic jamming systems[27,35–37], they are often limited to reconfigurability along few DOF due to the inherent mechanical complexity and integration[13,38]. Alternatively, compliant structures incorporated with architected stiffness-changing materials[17,21,22] could afford reconfigurability without increasing the devices' mechanical complexity.

[1]Morphing Matter Lab, Human-Computer Interaction Institute, Carnegie Mellon University, Pittsburgh, PA, USA. [2]Department of Mechanical Engineering, Carnegie Mellon University, Pittsburgh, PA, USA. [3]School of Design, Carnegie Mellon University, Pittsburgh, PA, USA. [4]Materials Science and Engineering, Carnegie Mellon University, Pittsburgh, PA, USA. [5]Mechanical and Industrial Engineering, University of Massachusetts, Amherst, MA, USA. [6]Department of Mechanical and Aerospace Engineering, Syracuse University, Syracuse, NY, USA. [7]BioInspired Syracuse, Syracuse University, Syracuse, NY, USA. [8]Mechanical Engineering, University of California, Berkeley, Berkeley, CA, USA. ✉e-mail: tzhang48@syr.edu; liningy@berkeley.edu

Although reconfigurable compliant mechanism designs affording binary modes have been explored[26,28], a design method targeting multimodal (>2 modes) reconfiguration is needed but not available. To address this, we adopt a screw algebra-based model[39–42] of conventional, non-reconfigurable compliant mechanisms, known as freedom and constraint topology (FACT), and extend it to account for multiple kinematic modes and reconfigurations. While such adaptation has been demonstrated in a recent study[19], it was a purely kinematic analysis and did not account for material properties, such as rod stiffness and buckling, in the design pipeline. Therefore, the previously designed devices can only reach a maximum stiffness of 0.79–10.2 Nmm⁻¹, navigating a much smaller space than the human kinesthetic perception range (0.013–59.342 Nmm⁻¹). Moreover, the prior work used passive flexures and tensioning cables for reconfiguration and had a maximum of 5DOF programmability (i.e., a minimum of one flexure is needed, adding 1 degree of constraint, denoted as DOC, to the system). In comparison, the active flexures used in this work allowed for 6DOF reconfigurability. On the other hand, we note that prior development in three-dimensional metamaterial and structures[43] had focused on actuation and proprioception along arbitrary DOF, and enabling kinematic and stiffness reconfiguration could further expand the design space.

In addition to the increased DOF of kinematic reconfigurability, our method also allows tailored design versatilities[44,45] for different use contexts[46–48] in terms of their stiffness ranges and form factors. In summary, we set the following criteria for designing compliant metastructures to extend the real-world implications of such devices: (i) the functions (kinematic freedoms and stiffness) should be actively reconfigurable to adapt to changing use contexts; (ii) the stiffness range should be tunable to accommodate target use cases. (iii) the devices should have customizable form factors for uses in different contexts (e.g., wear in target human body areas).

To achieve these design goals, we propose a compliant metastructure design (Fig. 1a) that is composed of both passive and active stiffness-changing flexural rods. Tailored design algorithms are presented that inform the topological arrangements, geometrical parameters, and control signals of these flexures based on target sets of reconfigurable kinematic modes. In this paper, we implemented systems that provide a large tunable stiffness changing ratio of up to 23.26x along a single DOF and a range of effective stiffness (2.445–73.785 Nmm⁻¹) tailored to the kinesthetic perception range (Fig. 1b).

To demonstrate the large and versatile design space of our approach (Fig. 1c), we implemented multiple wearable devices tailored to unique kinematic functions, body areas, and use contexts including using DOF reconfigurability to provide kinematic feedback when interacting with virtual reality[5] (Fig. 1d), simultaneously locking/unlocking multiple joints to provide targeted muscle group training[49] (Fig. 1e), context-adaptive rehabilitation and injury (e.g., carpal tunnel syndrome) prevention[50] (Fig. 1f), and proxying the haptic feelings of touching surfaces in mixed realities[19,51] (Fig. 1g).

## Results

### Mechanisms of reconfigurable compliant metastructure

Our fundamental design unit of the reconfigurable compliant metastructure consists of two rigid stages connected by parallel flexures (Fig. 1a and Supplementary Note 1). Multiple structural units can be serially connected to accommodate multiple motional freedoms at different locations (Fig. 1c). The flexures can be passive or actively stiffness-changing; their topological arrangements, geometrical factors, and control signals will determine the compliant metastructures' function and performance. Therefore, we used an algorithm-informed approach to design and control such metastructures.

We choose to engineer the actively stiffness-changing flexure with a resistive heating wire as the core and a thermoset epoxy resin-based cladding[52,53] (Fig.1a and Supplementary Note 1.5). A rod of 2 mm outer

diameter (OD) takes $31.45 \pm 2.58$ s to heat from the ambient temperature (25 °C) to its glass transition temperature of 54 °C and $67.90 \pm 4.95$ s to cool down. When heated, the resin's elastic modulus drops by 57 times from $1.14 \pm 0.18$ GPa to $0.02 \pm 0.008$ GPa. Due to their slender aspect ratio, in the cold state, both passive and active flexures have magnitudes higher stiffness against axial than bending or twisting loads, creating a degree of constraint along their axis. Yet, when the active flexures are softened, their stiffness and buckling loads are reduced proportionally to the elastic modulus, becoming soft and buckling easily against axial load. Therefore, stiffness-changing flexures can be used to create dynamic DOC, which in turn allows for kinematic reconfiguration.

Additionally, tactful flexure arrangements can instate distinct kinematic modes, each affording different mobilities and constraints (Fig. 2a). Each kinematic mode is defined by its DOF represented as a screw vector space [T] and a complementary DOC as a screw constraint space [W] (Fig. 1a and Supplementary Note 2). To instate a mode, the cold and stiff flexures should fully span the constraint space [W], which ensures the mode is exactly constrained without allowing motions not spanned by [T]. The passive flexures create a permanent constraint subspace shared by all modes, whereas the stiffness-changing flexures are used to dynamically expand or truncate constraint spaces. The resulting device can then be reconfigured between kinematic modes by selective softening and stiffening of active flexures.

### Rational design algorithm

We illustrate the steps to design kinematically reconfigurable devices (Fig. 2 and Supplementary Note 2.2), using the wrist joint device (Fig. 1f) as an example. This device aims at two kinematic modes where mode 1 enables flexion-extension and mode 2 enables ulnar deviation. Under a mode, any other freedoms except the one(s) enabled should remain constrained. Mode 1 can be represented by its freedom $[T_1]$ and constraint spaces $[W_1]$, and mode 2 by $[T_2]$ and $[W_2]$. The intersection of two modes' constraint spaces $[W_1] \cap [W_2]$ are shared by both modes. Consequently, flexures placed within this intersection are needed to exactly constrain both modes and are not required to be stiffness-changing. On the other hand, the relative complements $[W_1] \setminus [W_2]$ and $[W_2] \setminus [W_1]$ are the subspaces required to exactly constrain one mode but not the other, and flexures placed in this space should be stiffness-changing. Specifically, rods placed in $[W_1] \setminus [W_2]$ are needed to exactly constrain $[W_1]$ and would resist motions in $[T_2] \setminus [T_1]$, establishing mode 1. Conversely, to instate mode 2, the flexures residing in $[W_2] \setminus [W_1]$ should be hardened.

The screw subspaces parametrically describe the flexure placements that lead to the desired kinematic reconfigurability, allowing for rational and generative design and optimization toward design considerations (Fig. 2b, c). Notably, the constraint subspaces may also have redundancies or be invalid; therefore, only a subset (between $k$ and $2^k-1$, $k$ denotes the number of unique kinematic modes) of complement constraint subspaces is needed to create an exactly constrained device (Supplementary Note 2.6). The stages can be modeled into any shape that is sufficiently rigid (i.e., have minimal deflection) without altering the prescribed DOF modes[39], providing more design freedom for a customized fit or other functional purpose. In this work, we use an analytical stiffness model (Supplementary Note 3.1) to synthesize and adjust flexural rods' performance toward targeted values and leverage finite element (FE) simulation to verify the generated designs' performance (Supplementary Note 3.2).

To identify flexural configurations, the kinematic modes and flexures of a reconfigurable device can be represented as a Venn diagram (Fig. 2d), where each circle in the diagram represents the constraint space required to exactly constrain and instate a kinematic mode. The segments correspond to the subspaces resulting from the algorithm and, hence, flexures. To instate a kinematic mode (Fig. 2e), all flexures not included by the mode's constraint space should be softened while

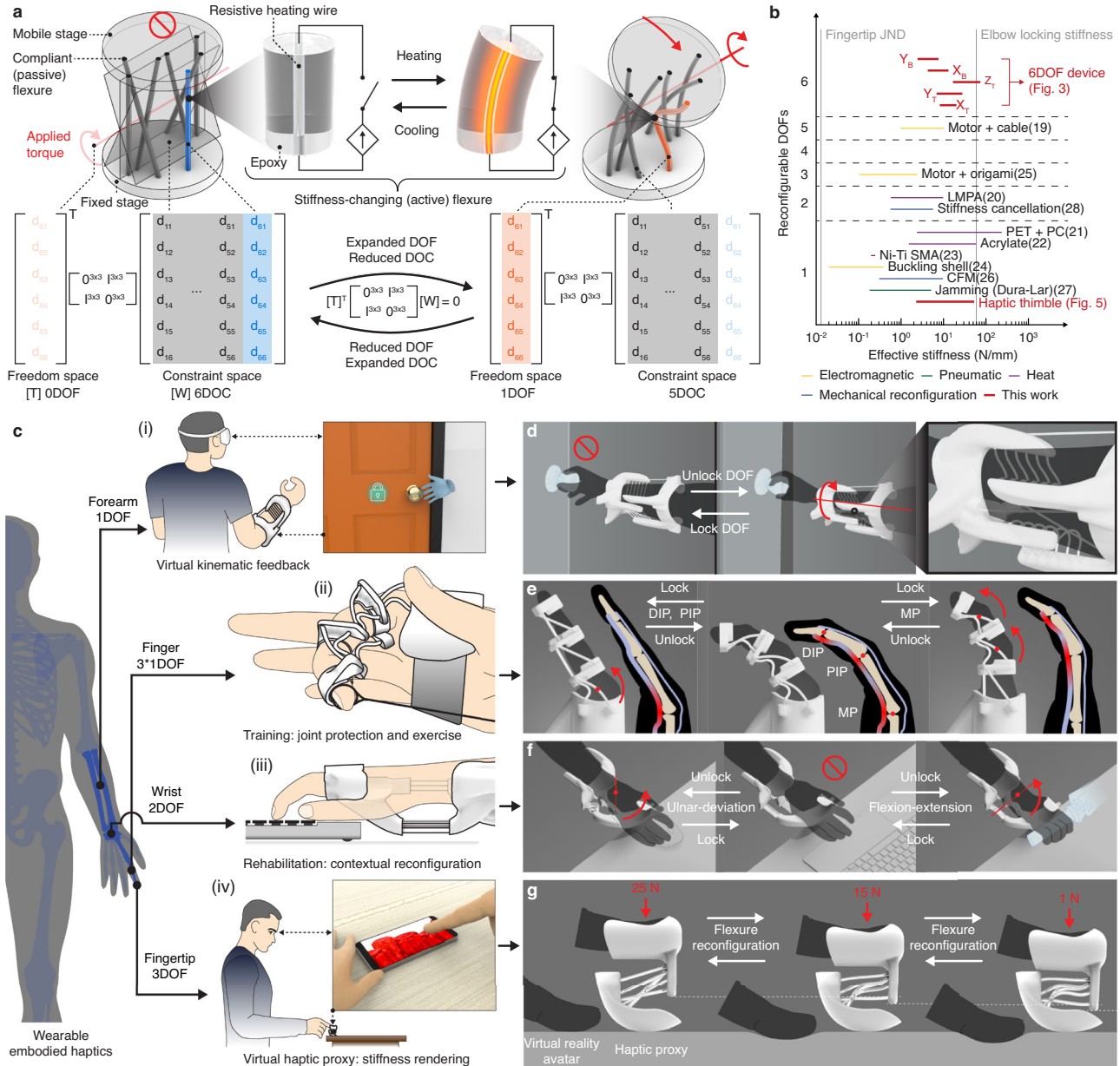

**Fig. 1 | Kinematically reconfigurable compliant metastructure design and envisioned application examples. a** Design of stiffness-changing materials for making compliant metastructures that can change their kinematics depending on the context of use and their screw algebra representation. DOF (degree of freedom), DOC (degree of constraint). **b** A benchmark of devices presented in this work and literature with respect to the number of programmable DOF and the range of afforded effective stiffness[19–28]. The range of stiffness needed for upper limb wearable kinesthetic haptic design is exemplified by the vertical dashed lines. JND (just noticeable difference), LMPA (low melting point alloy) [20], PET (polyethylene terephthalate)[21], PC (polycarbonate)[21], CFM (constant force mechanism)[26]. **c** Exemplary design space enabled by reconfigurable kinesthetic haptic device leveraging DOF locking/unlocking and stiffness changes, including (**i**) virtual kinematic feedback, (**ii**) selective muscle group training, (**iii**) context-adaptive rehabilitation braces, and (**iv**) wearable haptic proxies. **d** The device can disable the forearm's rotation to, e.g., simulate the experience of turning a locked vs. unlocked doorknob **e** The kinematic reconfiguration can selectively constrain finger interphalangeal joints, allowing for targeted muscle group training. MP (metacarpophalangeal) joint, DIP (distal interphalangeal) joint, PIP (proximal interphalangeal) joint. **f** The wrist device can function as a context-adaptive wrist brace (e.g., for alleviating wrist-tunnel syndrome) that can reconfigure its kinematic constraint to enable certain motions. **g** A haptic thimble device can proxy the haptic experience of pressing different materials by reconfiguring its stiffness.

the ones included should be kept stiff. In other words, the constraint subspaces (flexures) that are not encompassed by the mode's circle should be softened to lift their constraints, while the ones located within should be kept stiff to constrain unwanted mobilities.

In short, the rational design of reconfigurable kinematic devices can be summarized by the following five steps: (i) Compute each kinematic mode's kinematic freedom and constraint space; (ii) Compute the constraint space for placing non-stiffness-changing flexures; (iii) Compute constraint subspaces required to instate each mode; (iv)

Selecting constraint subspaces for placing stiffness-changing flexures and add non-redundant flexures required to exactly constrain each mode; (v) Adding additional flexures to reach the targeted device performance and modify the rigid stages to connect to the flexures and for other functions.

## Generalized design with 6-DOF reconfigurability

To exemplify the designs afforded by our algorithm, we employed it to create a device that can be reconfigured to provide any of the six

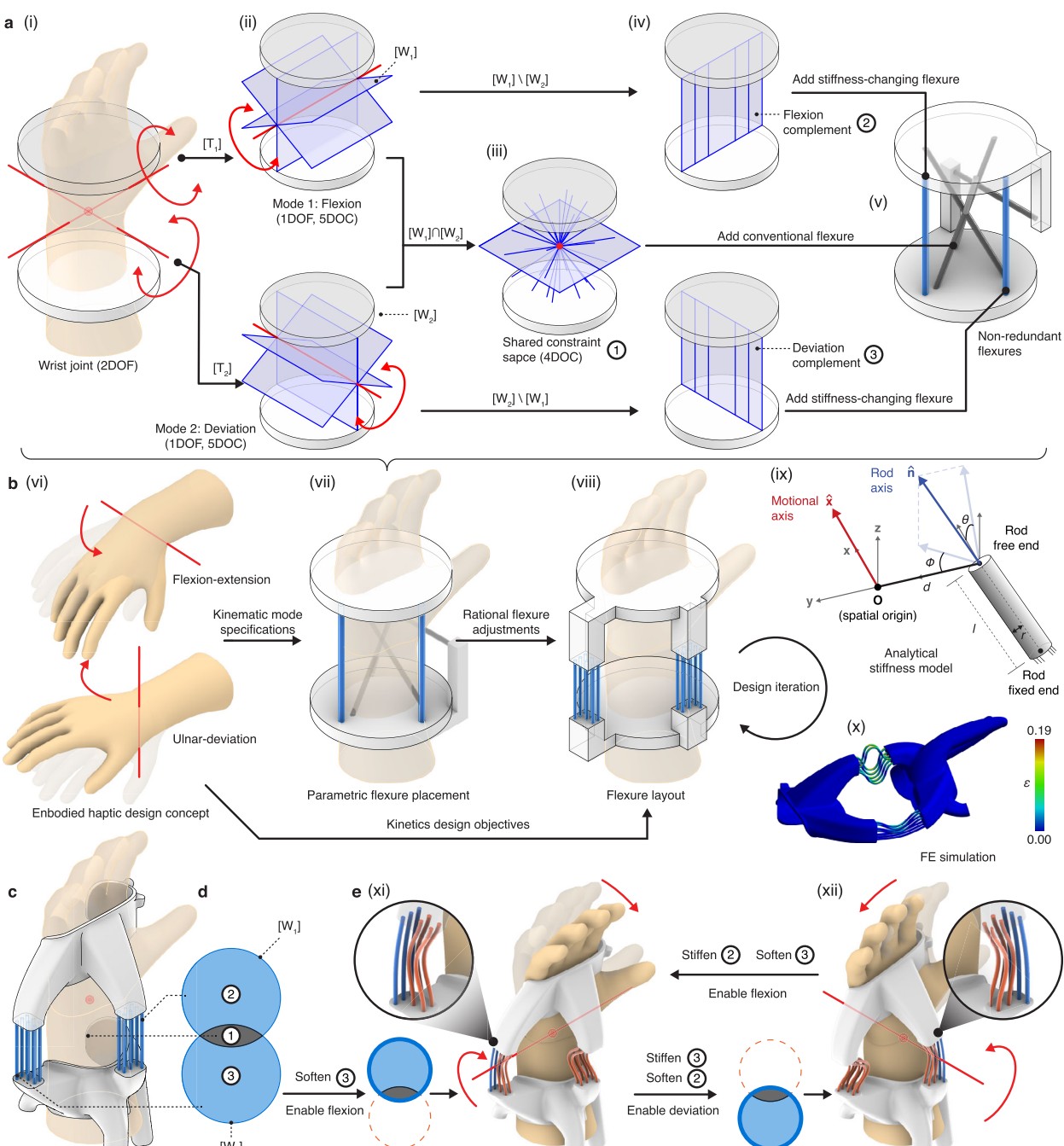

**Fig. 2 | Algorithm for designing kinematically reconfigurable compliant mechanisms using the wrist joint as an example. a** The algorithm starts by computing the (**i**) freedom and (**ii**) constraint space of deviation and flexion. (**iii**) The subspace for passive flexures is the intersection of all modal constraint spaces. In this design case, any rod flexure whose extended axis passes through the center point or lies on the plane spanned by the two degrees of freedom axes is permitted. (**iv**) The relative complements represent the rod placements required to exactly constrain the other mode. In this case, any flexure that does not pass through and is not parallel to the rotation axis is allowed. Next, minimal (**v**) non-redundant flexures should be added to span the constraint subspaces and exactly constrain each kinematic mode. In this case, four conventional flexures and two stiffness-changing flexures are used (one for each mode). DOF (degree of freedom), DOC (degree of constraint). **b** The overall design workflow for reconfigurable embodied haptic devices.

(**vi**) The input kinematic specifications are supplied to the algorithm (**a**) to find (**vii**) the parametric flexure placement. Based on the kinetics design goals (e.g., stiffness), (**viii**) more flexures could then be added to the design and use (**ix**) an analytical stiffness model and (**x**) finite element simulation to validate and iterate the design (color indicates the equivalent strain in ANSYS). In this sequence, the passive flexures from (**v**) are replaced by the wrist joint's skeleton to produce (**vii**), then more flexures are added, and their geometric parameters are altered to produce a device with the targeted kinetic performance. **c** The final design was created by remodeling the rigid stages in (**viii**) to provide a good fit to the wearer and connections between flexures. **d** The design's Venn diagram representation and the membership of each space in (**a**). **e** Reconfiguration for the kinematic modes: the complement constraint subspace should be canceled by softening the flexures to enable (**xi**) flexion and (**xii**) deviation.

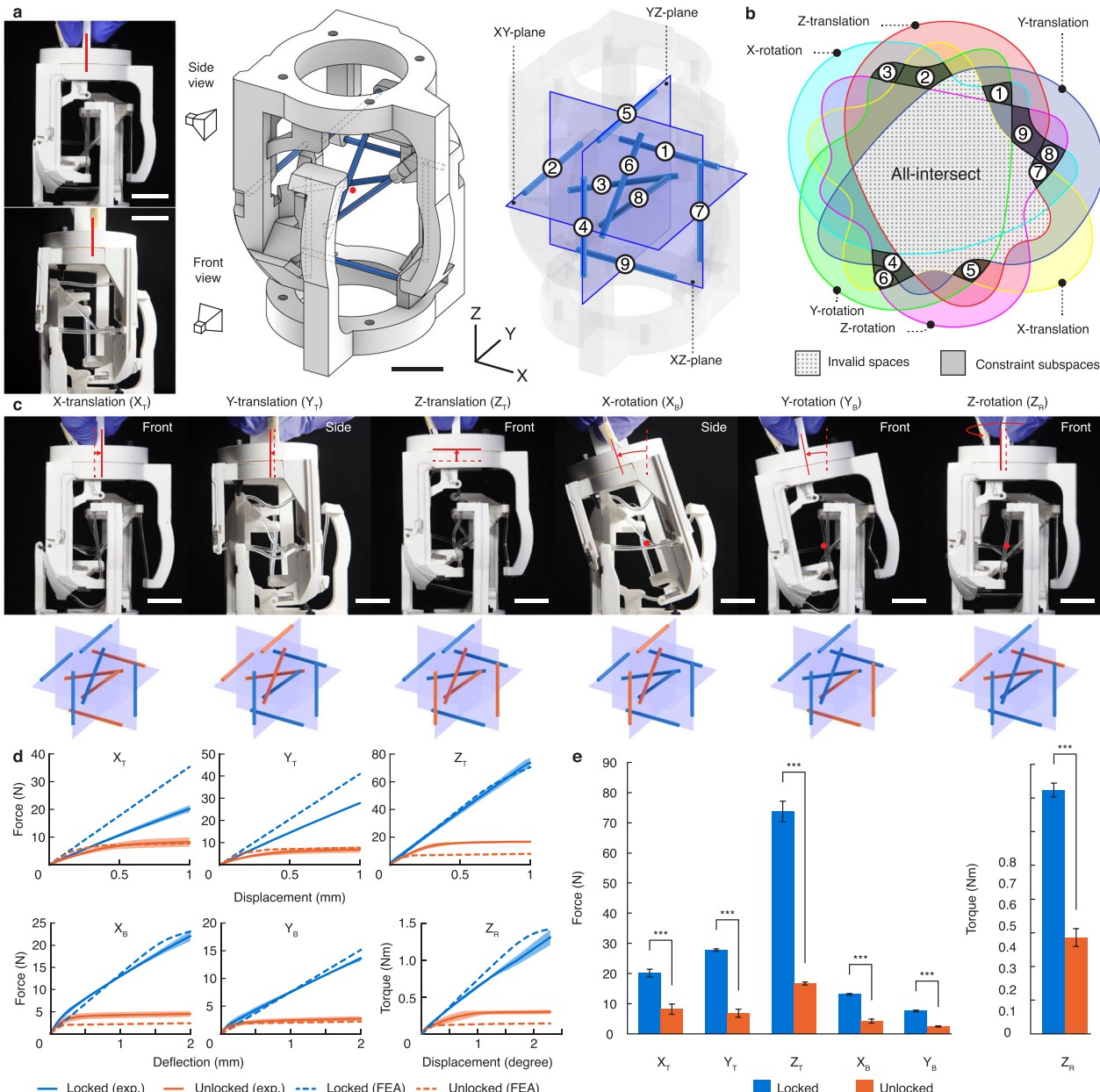

**Fig. 3 | A compliant mechanism joint that can be reconfigured to provide mobility along each and any of the six degrees of freedom (DOF) in the three-dimensional space. a** The device consists of two identical stages connected by nine stiffness-changing flexural rods lying on three orthogonal planes passing through the center of rotation (red dot). Scale bar, 20 mm. **b** The Venn diagram showing the constraint subspaces calculated from the algorithm and the affinity of each rod (Supplementary Note 2.5). **c** The device's motion along each of the DOF (top row) and the corresponding flexural rod configuration (bottom row, orange: heated, blue: cold). The dashed and solid lines show the mobile stage's centerline position before and after displacement, respectively, with the arrow showing the direction of motion. Scale bar, 10 mm. **d** The load-displacement plots of each DOF in the locked (blue) and unlocked (orange) states. Data are means ± s.d. $n = 3$ samples. **e** 1% (1 mm) displacement loads for each DOF. Statistically significant differences were found between toggled modes using $t$-tests (\*\*\*$p < 0.001$). Data are means ± s.d. $n = 3$ samples.

DOF in 3D space (Fig. 3a, Supplementary Note 4, and Supplementary Video S1). Each of the DOF was specified as a kinematic mode as input to the algorithm, leading to a total of $2^6 - 1 = 63$ constraint subspaces (Fig. 3b). Yet, several subspaces were invalid for being empty or led to unviable rod placements (Supplementary Note 4), leading to 56 viable subspaces to choose from for placing stiffness-changing rods. From this viable collection, we then picked nine subspaces that allowed us to place the rods on three planes through the targeted rotation center (Fig. 3a). The resulting device consists of nine stiffness-changing rods and zero passive flexures. The device has zero DOF when all flexures are stiffened. Yet, by softening the rods

according to the algorithm, the device can become mobile in each of the six DOF (Fig. 3c).

The device was jigged and tested to reveal its distinctive load-displacement behaviors between the locked and unlocked states along each DOF (Fig. 3d). The translational DOF were tested by linearly displacing the free end, whereas the X- and Y-rotational DOF were tested as bending deflections. Z-rotation was applied as a pure rotation by fixing the rotation center. Load curves were relatively linear for the locked states but displayed a plateau in the unlocked states. Consequently, the loads at the end of tests were also statistically distinct between the two states (Fig. 3e). The

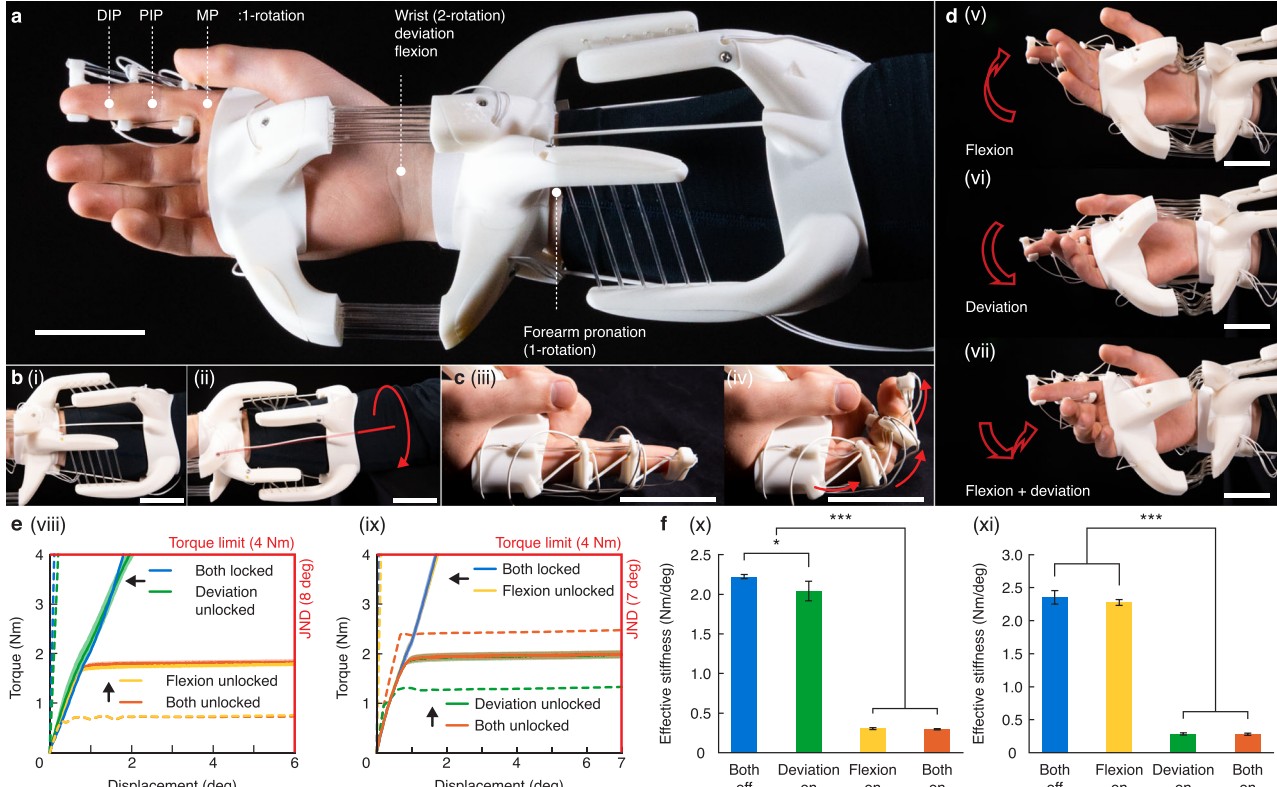

**Fig. 4 | Tailored design for wearable kinesthetic haptics. a** Picture of a device for the arm to toggle individual joint degree of freedom (DOF). MP (metacarpophalangeal) joint, DIP (distal interphalangeal) joint, PIP (proximal interphalangeal) joint. Scale bar, 50 mm. **b** Pictures of the unlocked forearm joint (**i**) before and (**ii**) after pronation. Scale bar, 50 mm. **c** Pictures of the unlocked finger joints (**iii**) before and (**iv**) after flexion. Scale bar, 50 mm. **d** Pictures of the wrist joint exercising along the unlocked (**v**) flexion, (**vi**) deviation, and (**vii**) both directions. Scale bar, 50 mm. **e** The load-displacement plots of the wrist joint device's (**viii**) flexion and (**ix**) deviation DOF under different configuration modes. JND (just noticeable difference). Data are means ± s.d. *n* = 3 repetitions. Dashed lines are finite element simulation results. **f** Stiffness comparison of the wrist joint device under different configuration modes against the (**x**) flexion and (**xi**) deviation DOF. The effective stiffness was calculated as load/displacement at the experiment criteria (i.e., torque limit load for locked states and JND limits for unlocked states). Data are means ± s.d. *n* = 3 repetitions.

largest difference was observed in the Z-translational (4.42x) and the smallest in the X-translational DOF (2.47x). We note that the difference is bound to become larger with increasing displacements as the unlocked state has a stiffness close to zero due to the buckling of flexures.

## Wearable kinesthetic haptic devices for mobility reconfigurations

The proposed design approach allowed us to tailor wearable devices for human augmentation. Here, it is demonstrated through the design of a device that can be worn on the arm and hand (Fig. 4a, Supplementary Note 5.1–5.5, and Supplementary Video S2, S3). The kinematics and performance were both considered per joint, and the joints were designed individually and combined later to form the complete device. In the rest pose, the arm and finger are fully extended. The joints were designed through the rational design algorithm and can be reconfigured to lock or unlock each of the afforded DOF. Note that when designing a wearable device, the skeletal structure can be considered a part of the passive flexures, which readily and exactly constrains the DOF. Therefore, it is optional to add passive flexures, though they may provide benefits such as maintaining the relative position between two stages.

In addition to kinematic modes, the device's stiffness with respect to the human body's performance and perception should also be considered for embodied haptics. We set our design criteria based on the torques exerted by each human body joint[54,55] and the just noticeable difference (JND) of joint angle proprioception[56] (Table S5).

To lock a DOF at a body joint, the device should displace less than the JND when subjected to the exertable torque, such that the wearer cannot perceive any movement. Conversely, in the unlocked state, the joint should be able to displace to and above the JND with a load lower than the exertable torque.

The forearm joint's pronation/supination is defined as an axial rotation along the length of the arm (Fig. 4b). The two stages are placed at the ends of the forearm and connected by twelve stiffness-changing flexures for reconfiguration and three passive flexures to maintain the spacing between stages. Based on the flexure placements suggested by the algorithm, we iteratively designed the device against the criteria (torque and JND) by changing the placement and number of rods in the system (Supplementary Note 5.4). Our FE simulation revealed that when locked, the device has a displacement of 0.83° under the torque limit of 5 Nm, much lower than the JND of 8°, suggesting that the device is perceptually immobile. Yet, when the joint is unlocked, the device requires only 0.79 Nm to reach the JND, and the device becomes compliant against the wearer's motions.

The three finger joints each has a rotational DOF about the interphalangeal joint and share an identical design (Fig. 4c). The rigid stages are added at the phalanges and are connected by a stiffness-changing flexure and a passive flexure, and the device was designed by changing the distance between the stiffness-changing flexures and the rotation axes. Similar to the forearm joint, our FE analysis also verified that the device is perceptually immobile in the locked state but becomes mobile upon heating the stiffness-changing flexures.

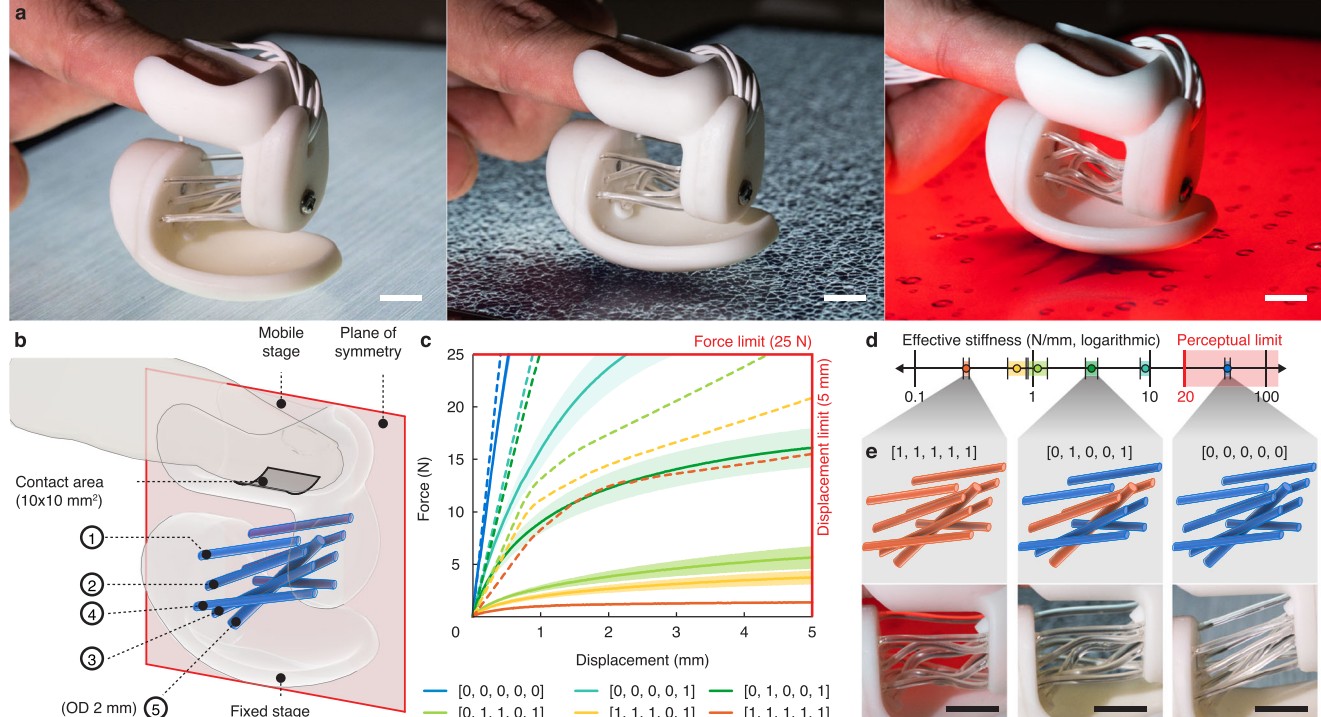

**Fig. 5 | A tailored design of embodied haptics proxy. a** Pictures (from left to right) of the haptic thimble simulating the stiffness of pressing on a block of aluminum stock, polyurethane foam, and a block of jelly. Scale bar, 10 mm. **b** The device design schema. All rods have an outer diameter (OD) of 1.5 mm unless specified otherwise. **c** The load-displacement plots of the thimble under different configurations. The five numbers making up the name of each sample indicate the states of flexure groups 1-5 in (**b**) with 1 indicating softening and 0 for hardening. Data are means ± s.d. $n = 3$ samples. Dashed lines are FE simulation results. **d** The device simulates different levels of stiffness within the design parameters, spanning two orders of magnitudes. Data are means ± s.d. $n = 3$ samples. **e** Flexure configurations (top row) and pictures of deformed flexures (bottom row) under three modes: [0, 0, 0, 0, 0] being fully rigid, [0, 1, 0, 0, 1] being partially softened, and [1, 1, 1, 1, 1] being fully softened. Scale bar, 10 mm.

Different from the finger and forearm joints affording single rotational freedom, the wrist joint allows rotation about two axes — flexion-extension and ulnar-radial deviation (Fig. 4d). Each rotation can be individually locked or unlocked, leading to four possible kinematic modes. The joint design consists of fourteen stiffness-changing flexures with three rods placed in-plane with each of the rotation axes. To enable a rotational freedom, all stiffness-changing flexures should be softened, except for those coplanar with the motional axis. Softening all flexures at the same time will allow both flexion and deviation to become unlocked. While the kinematics design of the wrist joints had been detailed in Fig. 2, the exact numbers of flexures were further determined through an iterative design process comparing the stiffness values from FE simulations with our design criteria, including the human wrist joint torque limit and JND.

We evaluated the wrist joint design through mechanical tests (Supplementary Note 5.3 and Supplementary Video S4). We isolated the device's wrist joint, mounted it on an articulated testing jig, and loaded it along each of the DOF to measure its responses (Fig. 4e and Supplementary Note 6.4). When both DOF were locked, the joint displaced by 1.81° ± 0.02° and 1.71° ± 0.07° at the torque limit when loaded with flexion and radial deviation, much lower than the JND of 8° and 7°, respectively. Effective DOF locking was observed when the other DOF was unlocked: the displacement along flexion was 1.76 ± 0.03° when radial deviation was enabled, indicating that the flexion DOF remained perceptually immobile. The same was also observed for the radial deviation DOF when flexion was unlocked (1.97 ± 0.12°). On the other hand, when both DOF were unlocked, the device required 1.82 ± 0.06 and 1.98 ± 0.09 Nm of torque to displace to the JND threshold along flexion and radial deviation, respectively, indicating the device could move past the JND with a torque lower than

the limit and was perceptually mobile. The effective stiffnesses calculated at the JND and torque limit intercepts (Fig. 4f) also showed no statistically significant differences ($p > 0.5$) between identical modes except for when flexion was locked ($p = 0.33$). Additionally, a strong statistical difference ($p < 0.001$) was observed between the locked and unlocked states, showing the device had distinctive perceived stiffness between locked and unlocked modes within human kinesthetic limits.

## Wearable haptic thimble device for stiffness tunability

In addition to mobility reconfiguration, the compliant metastructure design also enabled us to create a haptic device that renders a wide spectrum of stiffnesses. A device was designed to be worn on the fingertip to proxy the haptic feedback of pressing a surface to explore its material elasticity (Fig. 5a, Supplementary Note 5.6–5.9, and Supplementary Video S5). In such exploratory tasks, the finger applies forces across a 10 mm square area with a maximum force of 25 N, and the stiffness JND is 20 N/mm[57]. The device should be perceptually immobile when fully stiffened and could displace by up to 5 mm in the softest mode with a fraction of the maximum force.

The device was designed to have zero DOF in the stiffest and six DOF in the softest mode. Yet, while the kinematic modes are binary, we added stiffness-changing flexures with redundancy to create different levels of resistance against compression (Fig. 5b). Four pairs of stiffness-changing flexures (1.5 mm diameter) were added in mirror symmetry. The flexure pairs have slightly different orientations and positions. When selectively softened, the flexures create different levels of stiffness, buckling plateau, and hence haptic response. A 2 mm diameter stiffness-changing flexure was added to provide higher stiffness in the fully locked mode. Each group of rods can be heated or cooled separately, leading to $2^5$ reconfiguration modes.

We tested the device under a subset of configuration modes to find its afforded range of stiffnesses (Fig. 5c, d and Supplementary Video S6). The effective stiffness for locked and unlocked states (Fig. 5e) varied by 172.26 times from $0.27 \pm 0.02$ N/mm to $46.51 \pm 2.42$ N/mm, which were calculated at the JND and force limits, respectively. This range of stiffness corresponds to an effective modulus of 54.4 kPa to 10 MPa, considering the device's design parameters. Perceptually speaking, this range of elasticity is identical to the sensation of touching jelly and rubber. Yet, in the stiffest state, since the thimble's compliance is lower than the JND, the device is virtually undeformable to human perception and, therefore, can be used to proxy the hand feel of pressing stiffer materials, such as aluminum. Further repeatability tests also revealed the thimble device performed relatively consistently over 100 loading cycles (Supplementary Note 6.6).

## Discussion and Conclusion

In this work, we present a reconfigurable metastructure design concept, as well as a rational design pipeline that enables customizing devices targeted for different use contexts, kinematic reconfigurations, and stiffness demands. The design pipeline first employs an enhanced FACT design algorithm for calculating flexural rod placements given multiple reconfigurable kinematic modes. FE simulations and an analytical model are then used in conjunction to evaluate and iterate designs toward target kinematic performances. With this design strategy, we designed a generalized reconfigurable metastructure-based device that can selectively lock and unlock motions along any of the six DOF. Mechanical test results showed that the devices have statistically distinct load-displacement performances between different modes. A wearable device tailored for the arm, hand, and fingers is also provided to demonstrate our method's effectiveness at addressing kinesthetic demands, as well as the enabled design space. In particular, the devices can provide stiffness levels appropriate for wearable kinesthetic contexts. The stiffness change between the locked and unlocked states along a DOF is also sufficiently large, given human kinesthetic perceptual ranges. Finally, a haptic thimble exemplifies two magnitudes of stiffness change, showing the system's ability to render distinct haptic feedback.

While this work mainly focuses on the design of kinematically reconfigurable devices, the stiffness-changing rods also present opportunities for future improvements. Specifically, while the heating wires enable convenient electrothermal stiffness control, their low extensibility and high elastic modulus limited the rod's ability to conform to extension, causing the unlocked device to only allow motion along the rod-compressing direction. While bi-directional motions may be achieved by serially connecting two joints that enable the same DOF in opposite directions, we speculate that replacing the heating wires by modifying the epoxy cladding for electrothermal functions (e.g., Carbon nanotube fillers[58]) may also enable two-directional motion without further complicating the mechanical design. Since the metal wire has an elastic modulus that is magnitudes higher than that of the heated epoxy cladding, they contribute to most of a heated rod's stiffness. Removing the heating wire may also further amplify the rod's axial stiffness change between locked and unlocked states, therefore enabling more application opportunities. Still, epoxy's hysteresis (Supplementary Note 6.3) should be considered when designing for applications that require high precision, such as nano-positioning stages.

The stiffness-changing rods showed shape-memory effects that enabled them to recover from a cooled, deformed state when reheated to 54 °C (Supplementary Note 6.3). The degree of recovery depends on the type and magnitude of the deformation. A rod is relatively capable of recovering from bending (i.e., deformation along its DOF) but limited in recovering from buckling (i.e., deformation along its DOC). The difference likely results from the metal heating wire's plastic deformation, as literature[59] has reported that epoxy of similar composition

is capable of >96% shape recovery. Importantly, we note that the passive and unheated rods in a device design will provide additional recovery forces due to their springiness. In the wearable device case, the user may also adjust their limbs to help the rods and device recover to their rest pose. Future works may consider removing the heating wire and using an electrothermal epoxy to improve the rods' recoverability.

The proposed mechanism relies on joule heating and passive heat dissipation to change modes, thus requires longer reconfiguration time and is less suitable for applications that require frequent mode-switching (e.g., gaming). Still, use cases (Fig. 1c) that require occasional mode-switching (e.g., motor skill rehabilitation[60,61]) may still benefit from the proposed technique. Productivity applications (e.g., haptic material rendering or data visualization) may also tolerate waiting time to render haptic feedback. Nonetheless, future research may improve mode-switching time by fine-tuning the rod's heating and cooling profiles[62,63] or incorporating active thermoelectric components[64] to make the proposed method more versatile and generalizable.

The presented design principles can be generalized and applied to different body locations to provide kinesthetic feedback. The devices could also be digitally controlled and interfaced with a computer to provide kinesthetic experiences when interacting with virtual or augmented realities[51,65,66], on-demand body training and assistance[49], or just-in-time personal care by adapting their functions[38,50]. To scale up production, fabricating the entire device through a unified digital fabrication process (e.g., embedded printing of epoxy[67]) could help to reduce labor demands, assembly complexity, and navigate a larger design space. Combining the rational design pipeline with design optimization toward more diverse objectives such as size, weight, form factor, and energy consumption will also be important in future studies to make the devices more wearable, friendly to use, and resilient for daily life usage.

## Methods

### Device fabrication and control

The stiffness-changing rods are fabricated from a mixture of Epon 828 and Epikure 3380 at a 10:4 weight ratio. The mixture is injected into silicone tubes secured to an aluminum jig with heating wires (34-gauge 316 L stainless steel wire, Master Wire Supply) aligned at the center of the tubes. The cast rods are then left to gel at room temperature for 24 h, followed by curing in a 100 °C oven for 5 h. Once cured, the rod is left to cool down to room temperature, followed by removing the silicone tube to retrieve the stiffness-changing flexure rods. The details of the stiffness-changing flexure and device fabrication, assembly, and control are summarized in Supplementary Note 1. The fabricated rod's heating is controlled by sending a 0.4 A current through the wire with a DC power supply. All passive flexures and the 6DOF device's rigid stages are printed with the Ultimaker S5 printer with the Ultimaker white PLA filament. All other parts are printed with the Formlabs 3B printer with Formlabs' white resin V4. The fabrication details, heating control, and printer settings can be found in Supplementary Note 1.

### Rational design pipeline

The rational reconfigurable kinematics design algorithm is described in Supplementary Note 2. The analytical stiffness model and FE simulation setup are described in Supplementary Note 3. We use Ansys Mechanical to conduct the FE simulations using the Static Structural implicit solver. All digital models are created in Rhinoceros 3D 7, exported as STEP files, and imported into Ansys Mechanical for discretization and simulation.

### Device design

The 6DOF device design and design process are documented in Supplementary Note 4. The rational design workflows of the wearable device and haptic thimble are documented in Supplementary Note 5.

## Device characterization

The material characterization and mechanical experiment procedure are detailed in Supplementary Note 6. All mechanical tests are conducted with a Universal Testing Machine (UTM, Instron 5969) with custom fixtures to secure the samples while allowing certain DOF to remain mobile. The stiffness-changing epoxy's mechanical properties are characterized with the ASTM D412 test method, and its glass-transition temperature is determined with a Dynamic Mechanical Analyser (DMA, RSA–G2 Deta; TA Instruments). Images and videos are recorded with Sony (A7 III) and Canon (5D Mark II) DSLR cameras. Thermal images are recorded using a thermal imaging camera (HTi, HT-19). All images and videos are used without post-processing except for adjusting brightness and contrast for readability. Videos are composed using Adobe Premiere Pro.

## Data availability

The experiment data generated in this study are provided in the Supplementary Information. A copy of the data is also available at https://doi.org/10.5281/zenodo.14251242.

## Code availability

The code for the rational design algorithm and the analytical stiffness model is available upon request to H.Y. or L.Y.

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

## Acknowledgements

The authors acknowledge funding support from the U.S. National Science Foundation, including IIS-CAREER-2047912 (L.Y.), IIS-CAREER-2427455 (L.Y.), IIS-2118924 (L.Y.), and CMMI-2020476 (T.Z.), and the Translational Fellowship award from the Center for Machine Learning and Health at Carnegie Mellon University.

## Author contributions

H.Y. and L.Y. conceived the initial concept. H.Y. and G.O. formulated the reconfigurable kinematics design algorithm. H.Y., L.Y., T. J., and K.Z. conceived the application context. L.Y., T.Z., C.M., M.I. supervised the project. H.Y., L.Y., and T.Z. wrote the manuscript. H.Y., T.J., and K.Z. performed fabrication. H.Y., D.K.P., and K.Z. performed mechanical tests and material characterization. H.Y. and T.Z. performed mechanical modeling and simulation. L.Y. and T.Z. provided scientific and experimental advice. All authors commented on the manuscript.

## Competing interests

There is a patent related to this work filed by Carnegie Mellon University with the U.S. patent office (assignors: H.Y., M. I., C. M., L. Y., D.K.P., T. J., K. Z., G. O.; application no. PCT/US2023/015125). The other authors declare no competing interests.
