## [Transparent Peer Review file · Nature Communications]

A Compliant Metastructure Design with Reconfigurability up to Six Degrees of Freedom

Corresponding Author: Dr Lining Yao

Version 0:

Reviewer comments:

Reviewer #1

(Remarks to the Author)

This research introduces a novel concept of compliant meta-structures that can achieve customized degrees of freedom (DOF) and stiffness with the same device. The release and constraint of certain motions can be transferred from one to the other by heating the rods and changing their stiffness. This compliant meta-structure concept can be regarded as a dynamic application of the classical Freedom and Constraint Topology (FACT) method. It shows application values in fields such as rehabilitation, virtual realities, robotics, etc.

The following suggestions would help improve the quality of this paper:

- 1) The transformation process of the proposed devices from one kinematic mode to another involves the cooling of the stiffness-changing rods. How long does this cooling procedure take overall? In other words, what is the response time from the device to transfer from one mode to another? In addition, are there any specific requirements for this cooling procedure of the rods? If the rods are cooled when buckled, will they be able to return to their straight shapes automatically? Although relevant data is provided in the supplement sections, this information should be clarified in the main body of the manuscript, while it is not well-stated in the current version.
- 2) The authors developed a 6-DOF mechanism with 9 spatially arranged rods to demonstrate the effectiveness of the multimodal kinematics algorithm. Is the arrangement of the rods the only solution for achieving the desired 6-DOF mechanism? Is there an alternative spatial placement of the rods that would yield equivalent performance?
- 3) The authors also presented an arm-wearable kinesthetic haptic device, as shown in Figure 4. Naturally, a human wrist can bend to the left/right side, forward, and backward. However, it seems that the developed device, in its unlocked modes, halves the motion range of the wrist. As shown in Figure S27, backward and rightward bending motions of the wrist are not available due to the fixed length of the rods. This issue makes the developed arm-wearable device less attractive in practical applications. Are there any solutions to this problem?
- 4) The text font of the manuscript is confusing. For example, the font of the main text and the titles of figures should be different. Should the subsection title "Results" be in bold format?
- 5) In the sentence on lines 278 and 279, it should be "a stiffness-changing flexure and a passive flexure", or "stiffness-changing and passive flexures". Double-check the grammar throughout the paper to avoid such minor mistakes.
- 6) Will the heat-sensitive epoxy beams lengthen after being heated? If yes, would this influence the final performance of the devices in different kinematic modes?

Reviewer #2

(Remarks to the Author)

I co-reviewed this manuscript with one of the reviewers who provided the listed reports. This is part of the Nature

Communications initiative to facilitate training in peer review and to provide appropriate recognition for Early Career Researchers who co-review manuscripts.

Reviewer #3

(Remarks to the Author)

Title: Compliant Metastructure Reconfigurable at Six Degrees of Freedoms

This is an excellent paper about an important subject that has been studied and demonstrated with rigor and quality. If the editor deems the paper not on the caliber of Science, I believe the paper would be well suited for publication in Science Robotics. The paper is important because enabling compliant mechanisms where all 6 degrees of freedom (DOFs) can be selectively turned on and off via stiffness control is a kind of 'holy grail' in the field. The authors not only came up with a novel way to achieve this extraordinary ability using impressive advancements in kinematics but they also demonstrated the concept using multiple applications that are practical and important. The paper is also well written and the applications demonstrated have been thoroughly characterized.

I do have some concerns though that I would like the authors to address before I give my acceptance into a journal as renowned as Science:

- 1) My biggest concern with the concept is the time that it takes for the beams to change their stiffness. 30 seconds to heat up and become soft and then 3 minutes to cool down and harden seems like an unreasonably long time to wait for the proposed applications. Can the authors justify how this is acceptable for their chosen applications?
- 2) I imagine there would also be issues if the applications operated in unusually hot or fluctuating environments. How sensitive is their performance to fluctuations in ambient temperatures?
- 3) Does the flexure rod's material degrade over time as it is cyclically heated and cooled? If so how many heating and cooling cycles does it take for a difference to be noticed? I know the haptic device's repeatability was tested in Section 6.6, but I don't fully understand what was cycled. It seems that just the force was cycled (not heating and cooling?). It is also not clear over what range of deformation the device was cycled. Hopefully it was tested over the intended range of the application. It also looks like it was cycled in tension and compression given the positive and negative forces shown but the haptic example would only be loaded in compression? I suggest adding more detail in that section to explain what was done since repeatability is important to the concept proposed.
- 4) Also related to repeatability is hysteresis. It would be nice to characterize how much hysteresis the flexure rods experience as they are deformed over different cycles. I noticed the authors say the following at the beginning of section 6.4: "All tests were repeated for multiple cycles, and the second load cycles were used in the reported plots unless otherwise specified." Why was this the case? I imagine it is because the second cycle was more representative of the behavior of the rods since the first cycle of a material that exhibits high hysteresis is usually the most different from the latter cycles. I recommend sharing hysteresis loops and including the first one for a single rod to characterize how big a deal it is. Perhaps a reasonable amount of hysteresis is acceptable for the chosen applications but readers should know not to use the proposed approach for controlling stiffness if they wish to design high precision applications like nano-motion stages etc. Also given the current Instron testing setups shown, it may be difficult to distinguish the hysteresis generated by the devices and the hysteresis generated by the sliding-contact joints (e.g., the revolute joints) used in the custom-developed fixtures. You may need to design fixtures with flexure joints made of metal so that the fixtures avoid friction themselves so you can hone in on the hysteresis from your devices exclusively.
- 5) Although the flexure rods are intended to only change their stiffness in bending and compression, it may be useful and interesting to characterize how their stiffness may change for applications that require them to be loaded in tension. I suspect the stiffness would change at least a little bit. Your concept is so novel and applicable to constraint theory that it would be very useful to have such flexure rods fully characterized.
- 6) As for the characterization of the flexure rods in the paper, I'm confused by the plots in Fig. S34. In the text and in parts A-C it looks like they were compressed but in the plots of parts D-E they are labeled 'Tensile test of epoxy.' Shouldn't they be labeled "compression test of flexure rods." Also why is the error region so large for the OD 1.5 mm vs. the OD 2mm version? I might also recommend making a single plot for each OD dimension so the stiffness change from RT to 54C can be more easily identified instead of making a single plot for each temperature. I'd recommend providing more details in that section as well.
- 7) I'm guessing that S1 means sample 1, and S2 means sample 2, etc. in Fig. S33, but it doesn't seem to be stated anywhere in the paper. I'd write more to clarify the labels at least in the figure caption.
- 8) I'm impressed by how much the stiffness remains constant within the plots of Fig. 3D, but they don't seem to be plotted over a very significant range (1 mm seems very small compared to the size of the 6 DOF mechanism). I think the data should be plotted over the full intended range of the mechanism so readers can see if the stiffness starts drastically changing once the mechanism is deformed an appreciable amount. The concept is only great if it does maintain the desired stiffness for the most part over the full range.
- 9) I also noticed with these variable stiffness flexure rods that whether they are in a stiff or a compliant state, the initial undeformed stiffness remains unchanged and it requires a little motion to then cause them to buckle and have the stiffness drop for the compliant state. It is difficult to know how that feels for practical applications without playing with the mechanisms myself but I'm wondering if the authors can comment on that. It would of course be nice if when the rods were heated their instantaneous stiffness could also drop immediately before anything is even deformed a little bit.

Version 1:

Reviewer comments:

Reviewer #1

(Remarks to the Author)

The authors have mostly addressed my comments. I am happy to accept this paper in the current form.

Reviewer #2

(Remarks to the Author)

Reviewer #3

(Remarks to the Author)

The authors have sufficiently addressed my concerns. I believe the paper is ready for publication.

Reviewer #1:

This research introduces a novel concept of compliant meta-structures that can achieve customized degrees of freedom (DOF) and stiffness with the same device. The release and constraint of certain motions can be transferred from one to the other by heating the rods and changing their stiffness. This compliant meta-structure concept can be regarded as a dynamic application of the classical Freedom and Constraint Topology (FACT) method. It shows application values in fields such as rehabilitation, virtual realities, robotics, etc.

The following suggestions would help improve the quality of this paper:

Our response: We thank the reviewers for their positive comments and constructive feedback.

1. The transformation process of the proposed devices from one kinematic mode to another involves the cooling of the stiffness-changing rods. How long does this cooling procedure take overall? In other words, what is the response time from the device to transfer from one mode to another? In addition, are there any specific requirements for this cooling procedure of the rods? If the rods are cooled when buckled, will they be able to return to their straight shapes automatically? Although relevant data is provided in the supplement sections, this information should be clarified in the main body of the manuscript, while it is not well-stated in the current version.

Our response: We thank the reviewer for the insightful comments. The flexure rod takes 31.45 ± 2.58 seconds to heat from the ambient temperature (25°C) to 54°C and 67.90 ± 4.95 seconds to cool down. In the experiments, we allow devices to heat and cool for an extended period (three minutes) to allow their temperature to stabilize for consistent measurements.

We conducted new experiments to address the reviewer's comment on the shape recovery of the rods. It can be seen from Figure R1 that the rods showed shape-memory effects that enabled them to recover from a cooled, deformed state when reheated to 54°C . In addition, the degree of recovery depends on the type and magnitude of the deformation. The rod is relatively capable of recovering from bending (i.e., deformation along its DOF (degree of freedom)) but limited in recovering from buckling (i.e., deformation along its DOC (degree of constraint)). The difference likely results from the embedded metal heating wire's plastic deformation, as literature⁵⁹ has reported that epoxy of similar composition is capable of 96% shape recovery. Importantly, we note that the passive and unheated rods in a device design will provide additional recovery forces due to their springiness. In the wearable device case, the user may also adjust their limbs to help the rods and device recover to their rest pose. Future works may consider removing the heating wire and using an electrothermal epoxy formulation (e.g., using carbon nanotube fillers⁵⁸) to improve the epoxy rods' recoverability. We agree with the reviewer that more discussions are needed to clarify this point.

Figure R1 (Figure S38). Flexure rod recovery after bending and buckling.

Our modification to the manuscript: In Results: Mechanisms of Reconfigurable Compliant Metastructure (page no. 4 and line no. 119) and the supplementary materials (page no. xx and line no. 94, 98), we have updated the heating time description to “An OD 2 mm rod takes 34.45 ± 2.58 seconds to heat from the ambient temperature (25°C) to its glass transition temperature of 54°C and 67.90 ± 4.95 seconds to cool down.”

In Discussion and Conclusion (page no. 15 and line no. 379), we have added a paragraph about the shape memory effect and recovery:

“The stiffness-changing rods showed shape-memory effects that enabled them to recover from a cooled, deformed state when reheated to 54°C (Supplementary Note 6.3). The degree of recovery depends on the type and magnitude of the deformation. A rod is relatively capable of recovering from bending (i.e., deformation along its DOF) but limited in recovering from buckling (i.e., deformation along its DOC). The difference likely results from the metal heating wire’s plastic deformation, as literature⁵⁹ has reported that epoxy of similar composition is capable of $>96\%$ shape recovery. Importantly, we note that the passive and unheated rods in a device design will provide additional recovery forces due to their springiness. In the wearable device case, the user may also adjust their limbs to help the rods and device recover to their rest pose. Future works may consider removing the heating wire and using an electrothermal epoxy to improve the epoxy rods’ recoverability.”

We also added Figure R1 (Figure S38) to Supplementary Note 6.3 (page no. 57 and line no. 993) and added a paragraph describing the observed shape-memory effect:

“Figure S38 shows a rod’s (OD 2mm, 50 mm long) shape memory effect that allowed it to recover from deformation. When deformed along its degree of freedom (i.e., 10 mm lateral deflection), a rod may recover most of the deformation without external loads. However, when subjected to deformation along its degree of constraint (i.e., 5 mm compression), the rod was unable to completely recover from buckling.”

2. The authors developed a 6-DOF mechanism with 9 spatially arranged rods to demonstrate the effectiveness of the multimodal kinematics algorithm. Is the arrangement of the rods the only solution

for achieving the desired 6-DOF mechanism? Is there an alternative spatial placement of the rods that would yield equivalent performance?

Our response: We thank the reviewer for the insightful comments. The rod arrangement demonstrated in Figure 3, consisting of 9 rods, is not the only solution for achieving a 6-DOF mechanism. We have added an alternate 6-DOF device design (Fig. R2) in the supplementary materials to illustrate this opportunity. In Supplementary Note 5, the design iteration histories also provide examples of varying flexure layouts that may satisfy a target kinematic function.

Figure R2 (Figure S17). Alternative 6-DOF device design. *a*, Device design and reconfigurable flexure rod layout. The red dot shows the pivot center. Scale bar, 20 mm. *b*, The Venn diagram showing each rod's constraint subspaces membership. *c*, Illustrations of the device's flexure configuration to enable each DOF (orange: heated, blue: cold).

Our modification to the manuscript: Added a paragraph and Figure R2 to Supplementary Note 4.2 (page no. 44 and line no. 731):

“We note that it is possible to use the rational design algorithm to produce varying designs for a given kinematic reconfiguration. Figure S17 shows an alternate 6-DOF device design with a different flexure layout. The algorithm solves for viable and needed constraint subspaces and flexure placements, and mechanism designers may interpret its output and employ different heuristics in modeling the flexural elements (as noted in Supplementary Note 2). A targeted reconfigurable mechanism may be satisfied by multiple constraint subspace combinations. Compared with the device shown in Figure 3, the device in Figure S17 requires more rods to create the required reconfiguration layout, and the rods are more cluttered.”

3. The authors also presented an arm-wearable kinesthetic haptic device, as shown in Figure 4. Naturally, a human wrist can bend to the left/right side, forward, and backward. However, it seems that the developed device, in its unlocked modes, halves the motion range of the wrist. As shown in Figure S27, backward and rightward bending motions of the wrist are not available due to the fixed length of the rods. This issue makes the developed arm-wearable device less attractive in practical applications. Are there any solutions to this problem?

Our response: We thank the reviewer for the insightful comments. The devices can only deform in the direction that compresses the rods due to the presence of heating wires. While the heated epoxy cladding can afford 30% of tensile strain before breaking (Fig. S33), the metal wires are stiff in tension, and the modulus mismatch between the wire and epoxy will cause the two to delaminate and slide against each other, compromising the device's mechanical properties. Two design strategies may overcome this problem. Designers may serially connect two joints, each allowing displacements along the same DOF in opposite directions. This way, the free end of the serial joints will have bi-directional mobility along a DOF when unlocked. Alternatively, replacing the heating metal wire with electrothermal fillers⁵⁸ may also improve the stiffness-changing rods' extensibility, therefore allowing two-directional motions when heated. We have added this information to the Discussion and Conclusion.

Our modification to the manuscript: Added a paragraph to Discussion and Conclusion (page no. 14 and line no. 365):

“While this work mainly focused on the design of kinematically reconfigurable devices, the stiffness-changing rods also present opportunities for future improvements. Specifically, while the heating wires enabled convenient electrothermal stiffness control, their low extensibility and high elastic modulus limited the rod's ability to conform to extension, causing the unlocked device to only allow motion along the rod-compressing direction. While bi-directional motions may be achieved by serially connecting two joints that enable the same DOF in opposite directions, we speculate that replacing the heating wires by modifying the epoxy cladding for electrothermal functions (e.g., Carbon nanotube fillers⁵⁸) may also enable two-directional motion without further complicating the mechanical design.”

4. The text font of the manuscript is confusing. For example, the font of the main text and the titles of figures should be different. Should the subsection title “Results” be in bold format?

Our response: We thank the reviewer for pointing this out. We have re-formatted the document to resolve this issue. All figure captions now use a font different from the main text.

5. In the sentence on lines 278 and 279, it should be “a stiffness-changing flexure and a passive flexure”, or “stiffness-changing and passive flexures”. Double-check the grammar throughout the paper to avoid such minor mistakes.

Our response: We thank the reviewer for pointing this out. We have proofread the document to correct grammar mistakes.

6. Will the heat-sensitive epoxy beams lengthen after being heated? If yes, would this influence the final performance of the devices in different kinematic modes?

Our response: We have conducted new experiments to verify the length change of the epoxy rods during heating. In the tests, we mounted an epoxy rod (50 mm long, OD 2 mm) on a rigid fixture to observe its deformations during a heating and cooling cycle (see the following image from left to right). The two ends are fixed, and if the rod lengthens after being heated, the length change should induce the rod to buckle or the rod to displace sideways. In the experiment, we did not observe any noticeable difference in the rod's geometry, showing that the rod did not lengthen by a significant amount when heated. Still, given epoxy's coefficient of thermal expansion⁸¹ of 54.8 ppm/°C, the rod should lengthen by $29 \times 50 \times 54.8 \times 10^{-6} = 0.079$ mm from room ambient condition to 54°C.

Figure R3. Image of flexure rods before and after heating at 54°C.

Our modification to the manuscript: None.

Reviewer #2

I co-reviewed this manuscript with one of the reviewers who provided the listed reports. This is part of the Nature Communications initiative to facilitate training in peer review and to provide appropriate recognition for Early Career Researchers who co-review manuscripts

Our response: We thank the reviewers for their constructive feedback.

Reviewer #3:

This is an excellent paper about an important subject that has been studied and demonstrated with rigor and quality. The paper is important because enabling compliant mechanisms where all 6 degrees of freedom (DOFs) can be selectively turned on and off via stiffness control is a kind of 'holy grail' in the field. The authors not only came up with a novel way to achieve this extraordinary ability using impressive advancements in kinematics but they also demonstrated the concept using multiple applications that are practical and important. The paper is also well written and the applications demonstrated have been thoroughly characterized.

Our response: We thank the reviewers for their positive comments and constructive feedback.

I do have some concerns though that I would like the authors to address before I give my acceptance into a journal as renown as Science:

1. My biggest concern with the concept is the time that it takes for the beams to change their stiffness. 30 seconds to heat up and become soft and then 3 minutes to cool down and harden seems like an unreasonably long time to wait for the proposed applications. Can the authors justify how this is acceptable for their chosen applications?

Our response: We thank the reviewer for the insightful comments. We agree with the reviewer that reconfiguration speed is a critical factor of the proposed physical system. We envision the proposed approach could be helpful in applications that require occasional mode-switching speed, such as the ones illustrated in Figure 1c. In particular, rehabilitation⁶¹ and training⁶⁰ often do not require frequent device mode-switching; training sessions are often divided into 5- to 10-minute segments that target specific joints, and reconfiguration can take place when the user is resting. Productivity VR applications, such as haptic material rendering or data visualization, may also tolerate waiting time to render feedback. On the other hand, we also acknowledge that the devices take around 30 to 60 seconds to change kinematic modes, making them less suitable for applications that require high reconfiguration frequency (e.g., gaming). Improving mode-switching time through fine-tuning electrothermal profile and active cooling is a very important and critical future research direction to make the proposed method more versatile and generalizable.

Our modification to the manuscript: We have added a paragraph in the Discussion and Conclusion (page no. 15 and line no. 390):

“The proposed mechanism relies on joule heating and passive heat dissipation to change modes, thus requires longer reconfiguration time and is less suitable for applications that require frequent mode-switching (e.g., gaming). Still, use cases (Fig. 1c) that require occasional mode-switching (e.g., motor skill rehabilitation^{60, 61}) may still benefit from the proposed technique. Productivity applications (e.g., haptic material rendering or data visualization) may also tolerate waiting time to render haptic feedback. Nonetheless, future research may improve mode-switching time by fine-tuning the rod’s heating profile^{62, 63} or incorporating active thermoelectric components⁶⁴ to make the proposed method more versatile and generalizable.”

2. I imagine there would also be issues if the applications operated in unusually hot or fluctuating environments. How sensitive is their performance to fluctuations in ambient temperatures?

Our response: We thank the reviewer for the insightful comments. Our original Figure S32 (now Fig. S33) shows the epoxy’s mechanical properties as a function of temperature. The tan delta remains more or less constant from 30°C to 40°C. From the data, the modulus of the epoxy rod will not significantly change (1.4% in tan delta) even at the temperature of 40°C. On the other hand, the heating and cooling time is also reduced with higher ambient temperature due to the smaller range of temperature change to reach T_g. We have added a paragraph to address this.

Our modification to the manuscript: We have added a paragraph to Supplementary Note 1.4: Heating Control (page no. 5 and line no. 112):

“The epoxy rods’ mechanical property is a function of temperature (see also Section 6.1). The environmental temperature mainly affects the rods’ stiffness under the unheated state. The higher the ambient temperature, the softer the epoxy rod becomes in the unheated state. Consequently, a rod would exhibit a smaller stiffness change between the two states. The epoxy’s tan delta remains more or less constant between 30°C to 40°C, and the moduli are still magnitudes higher than that in the heated state and sufficient to lock a DOF. Literature has shown that epoxy’s glass transition temperature⁵⁹ (T_g) and stiffness⁶⁸ can be tuned by altering the crosslinker ratio: the higher the crosslinker concentration, the higher the T_g and stiffness. When designing devices for unusually hot environments, using a higher crosslinker ratio may help to create a desired stiffness change and avoid undesired softening due to temperature fluctuations.

On the other hand, heating and cooling time may also be affected by the ambient temperature. At 25°C, a rod takes 31.45 ± 2.58 seconds to heat up to 54°C and 67.90 ± 4.95 seconds to cool down. At 30°C, the heating and cooling time are reduced to 29.12 ± 2.39 seconds and 40.72 ± 3.01 seconds, respectively. At 30°C, the heating is reduced to 25.62 ± 2.18 seconds, and the cooling time is lowered to 26.31 ± 2.06 seconds.”

3. Does the flexure rod’s material degrade over time as it is cyclically heated and cooled? If so how many heating and cooling cycles does it take for a difference to be noticed? I know the haptic device’s repeatability was tested in Section 6.6, but I don’t fully understand what was cycled. It seems that just the force was cycled (not heating and cooling?). It is also not clear over what range of deformation the device was cycled. Hopefully it was tested over the intended range of the application. It also looks like it was cycled in tension and compression given the positive and negative forces shown but the haptic example would only be loaded in compression? I suggest adding more detail in that section to explain what was done since repeatability is important to the concept proposed.

Our response: We thank the reviewer for the insightful comments. We have clarified the test conditions in the manuscript. In the original manuscript, only the load was cycled, and the device was tested over the intended range of the application. The negative values are a result of hysteresis, and we have applied baseline correction in this revision to account for its effect, making all measurements positive (Figure R4). We have also included a new experiment that tests a rod’s mechanical response during cyclic heating and cooling (Figure R5).

Our modification to the manuscript: Added descriptions of the test setup to Supplementary Note 6.6: Device Repeatability (page no. 65 and line no. 1112):

“The device was configured into three modes (fully locked [0, 0, 0, 0, 0], partially unlocked [0, 1, 0, 0, 1], and fully unlocked [1, 1, 1, 1, 1]) and displacement-loaded (0.6 mm, 5 mm, and 5mm, respectively) to measure their response across 100 cycles. The device was not heating/cooling-cycled between repetitions.”

Figure R4 (Figure S47). The haptic thimble repeatability test over 100 loading cycles.

We have also added a paragraph and figure about a cyclic heating/cooling test of a single rod (using the same setup as Fig. S34a, now Fig. S35a) to Supplementary Note 6.6: Device Repeatability (page no. 65 and line no. 1123):

“On the other hand, we have also added a cyclic cooling and heating test to study an epoxy rod’s performance degradation. In this test, a rod (OD 2 mm) was repeatedly cycled between RT and 54°C. In each cycle, the rod was heated or cooled for three minutes and then compressed by 1% for four cycles to measure its response. The following figure shows the measurements of a single rod’s mechanical response under 1% compression over ten heating and cooling cycles. There was no visible performance drop or trend in both heated and unheated states over 10 cycles.”

Figure R5 (Figure S48). An OD 2 mm rod’s compression test over ten cycles. The rod is conditioned at ambient temperature (blue) at odd number cycles and 54°C (orange) at even number cycles.

4. Also related to repeatability is hysteresis. It would be nice to characterize how much hysteresis the flexure rods experience as they are deformed over different cycles. I noticed the authors say the following at the beginning of section 6.4: “All tests were repeated for multiple cycles, and the second load cycles were used in the reported plots unless otherwise specified.” Why was this the case? I imagine it is because the second cycle was more representative of the behavior of the rods since the first cycle of a material that exhibits high hysteresis is usually the most different from the latter cycles. I recommend sharing hysteresis loops and including the first one for a single rod to characterize how big a deal it is. Perhaps a reasonable amount of hysteresis is acceptable for the chosen applications but readers should know not to use the proposed approach for controlling stiffness if they wish to design high precision applications like nano-motion stages etc. Also given the current Instron testing setups shown, it may be difficult to distinguish the hysteresis generated by the devices and the hysteresis generated by the sliding-contact joints (e.g., the revolute joints) used in the custom-developed fixtures. You may need to design fixtures with flexure joints made of metal so that the fixtures avoid friction themselves so you can hone in on the hysteresis from your devices exclusively.

Our response: We thank and agree with the reviewer for the insightful comments. We observed the first loading cycle to have higher hysteresis than the subsequent cycles and chose to report mechanical test results based on the second cycle as it is more representative of the device’s behavior in a dynamic situation. We have added a paragraph and figure to the Supplementary Note to discuss the observed hysteresis, as well as a sentence in the Discussion and Conclusion to inform their implications in high-precision use cases. The figure data (Figure R6) shows the 5-cycle load curve of a characteristic rod that appeared in Figure S34 (now Figure S35).

In the Instron tests, the jigs did not pose slide-contact joints, and we used metallic bearings to minimize friction between moving parts. This is now clarified in the manuscript.

Our modification to the manuscript: Added a paragraph to address hysteresis in Supplementary Note 6.3: Flexural rod characterization (page no. 57 and line no. 984):

“Figure S37 shows a characteristic OD 2 mm rod’s hysteresis under cyclic compression. At room temperature, the rod had a 15.67% hysteresis ratio in the first loading cycle, which subsequently dropped to 10.56% in the second and 9.51% in the fifth cycle. Similarly, at 54 °C, the initial loading cycle has the highest hysteresis of 38.33%, which then decreased to 18.64% and 16.91% in the second and fifth loading cycles, respectively.”

Figure R6 (Figure S37). Characteristic flexural rod hysteresis plots. a, The rod under room temperature. b, The rod under 54°C. The faded-out curves show intermediate cycles between cycle 1 and cycle 5.

Added a sentence in the Discussion and Conclusion (page no. 15 and line no. 376) to inform the use of epoxy rods in high-precision applications:

“Still, epoxy’s hysteresis (Supplementary Note 6.3) should be considered when designing for applications that require high precision, such as nano-positioning stages.”

Added a paragraph to address testing jig articulations in Supplementary Note 6.5: Mechanical Testing Jig Design (page no. 61 and line no. 1048):

“In the Instron tests, we designed the jigs without slide-contact joints. The parts are either clamped or fixed to the Instron machine or mechanically articulated. Each rotational axis annotated in Figures S41 through S45 is implemented with a metallic bearing between moving parts to minimize friction and avoid confounding measurements.”

5. Although the flexure rods are intended to only change their stiffness in bending and compression, it may be useful and interesting to characterize how their stiffness may change for applications that require them to be loaded in tension. I suspect the stiffness would change at least a little bit. Your concept is so novel and applicable to constraint theory that it would be very useful to have such flexure rods fully characterized.

Our response: We thank the reviewer for the insightful comments. We have added a new experiment to characterize the rods’ mechanical response under tensile loads. There is indeed a stiffness change between heated and unheated states. We have added a data figure (Fig. R7) and a paragraph to address this.

Our modification to the manuscript: Added a paragraph and figure to Supplementary Note 6.3: Flexural rod characterization (page no. 56 and line no. 973):

“Under both RT and 54 °C, a 50 mm long, OD 2 mm rod exhibited a relatively linear load-displacement curve when stretched due to the lack of buckling plateaus. The forces required to stretch the rod by 1% strain varied by 6.71 times, changing from 36.60 ± 1.07 N at RT to 5.45 ± 0.94 N at 54 °C. In comparison, the forces required to compress the rod by 1% varied 38.9 times, changing from 35.79 ± 2.14 N at RT to 0.92 ± 0.08 N at 54 °C.”

Figure R7 (Figure S36). Flexural rod tensile characterization (50 mm long, OD 2 mm). Data are means \pm s.d. $n = 3$ samples.

6. As for the characterization of the flexure rods in the paper, I'm confused by the plots in Fig. S34. In the text and in parts A-C it looks like they were compressed but in the plots of parts D-E they are labeled 'Tensile test of epoxy.' Shouldn't they be labeled "compression test of flexure rods." Also why is the error region so large for the OD 1.5 mm vs. the OD 2mm version? I might also recommend making a single plot for each OD dimension so the stiffness change from RT to 54C can be more easily identified instead of a making a single plot for each temperature. I'd recommend providing more details in that section as well.

Our response: This issue had been corrected in the Nature Communication submission. Figure S34 (now Figure S35, page no. 56 and line no. 966) now shows a plot for each rod diameter, and the labels have been corrected to "Compression test of epoxy rod."

Our modification to the manuscript: None.

7. I'm guessing that S1 means sample 1, and S2 means sample 2, etc. in Fig. S33, but it doesn't seem to be stated anywhere in the paper. I'd write more to clarify the labels at least in the figure caption.

Our response: Similar to the previous comment, this was also corrected in the Nature Communication submission. Figure S33's (Now Figure S34, page no. 56 and line no. 955) caption now explains S1 through S3 strands for samples 1 to 3 in the experiments.

Our modification to the manuscript: None.

8. I'm impressed by how much the stiffness remains constant within the plots of Fig. 3D, but they don't seem to be plotted over a very significant range (1 mm seems very small compared to the size of the 6 DOF mechanism). I think the data should be plotted over the full intended range of the mechanism so readers can see if the stiffness starts drastically changing once the mechanism is deformed an appreciable amount. The concept is only great if it does maintain the desired stiffness for the most part over the full range.

Our response: We thank the reviewer for the comments. In this design example, we aimed to demonstrate the design method's generalizability and ability to tackle complex kinematic reconfigurations (6-DOF). Hence, there was no specific design context and intended range of motion. Still, when testing the device, we chose a loading range similar to that used in the literature^{5, 6, 7, 10} (1% of device length) to create a common baseline for comparing stiffness reconfiguration.

Our modification to the manuscript: None.

9. I also noticed with these variable stiffness flexure rods that whether they are in a stiff or a compliant state, the initial undeformed stiffness remains unchanged and it requires a little motion to then cause them to buckle and have the stiffness drop for the compliant state. It is difficult to know how that feels for practical applications without playing with the mechanisms myself but I'm wondering if the authors

can comment on that. It would of course be nice if when the rods were heated their instantaneous stiffness could also drop immediately before anything is even deformed a little bit.

Our response: We thank the reviewer for the insightful comments. The linear region of the unlocked state peaked shy of 2Nm, which corresponds to 25% of the wrist's isometric strength. This load is similar to flexing one's wrist while holding a 2.5 kg object in the hand. In our experiences interacting with the device, there was indeed some resistance against motions in the heated state, but it did not significantly hinder the wearer's (a 22-year-old, healthy biological male) motions. The resistance also becomes relatively constant past 1 degree, the buckling threshold (Fig. 4e).

We fully agree with the reviewer that it would be nicer to decrease the initial resistant force further. We speculate that the initial linear region observed in the unlocked mode may have come from the heating wires. In this region, the stiffness is dominated by the rod's axial stiffness. While a wire (OD 0.15 mm) only occupies 0.6% of the rod's cross-section area, its elastic modulus (200 GPa) is two and four magnitudes higher than the epoxy cladding in the unheated and heated states, respectively. Consequently, the wires contribute 50.7% and 97.53% of the rod's axial stiffness in either state prior to buckling. We acknowledge that this is a limitation in the current stiffness-changing rod design, and future research may consider replacing the heating wire with other electrothermal mechanisms to minimize the rods' unlocked axial stiffness. In that case, the design algorithm still holds, and the resulting device will be able to achieve a more drastic resistance change between locked and unlocked states. We have added this comment to the manuscript.

Our modification to the manuscript: We have added a comment to the Discussion and Conclusion (page no. 15 and line no. 373):

“Since the metal wire has an elastic modulus that is magnitudes higher than that of the heated epoxy cladding, they contribute to most of a heated rod's stiffness. Removing the heating wire may also further amplify the rod's axial stiffness change between locked and unlocked states, therefore enabling more application opportunities.”

References

58. Farcas, C. *et al.* Ice-Prevention and De-Icing Capacity of Epoxy Resin Filled with Hybrid Carbon-Nanostructured Forms: Self-Heating by Joule Effect. *Nanomaterials* **11**, (2021).
59. Zhong, K. *et al.* EpoMemory: Multi-State Shape Memory for Programmable Morphing Interfaces. in *Proceedings of the 2023 CHI Conference on Human Factors in Computing Systems* (Association for Computing Machinery, New York, NY, USA, 2023). doi:10.1145/3544548.3580638.
60. Feng, H. *et al.* Virtual Reality Rehabilitation Versus Conventional Physical Therapy for Improving Balance and Gait in Parkinson’s Disease Patients: A Randomized Controlled Trial. *Medical Science Monitor* **25**, 4186–4192 (2019).
61. Merians, A. S. *et al.* Virtual Reality–Augmented Rehabilitation for Patients Following Stroke. *Phys Ther* **82**, 898–915 (2002).
62. Zadan, M. *et al.* Liquid Crystal Elastomer with Integrated Soft Thermoelectrics for Shape Memory Actuation and Energy Harvesting. *Advanced Materials* **34**, 2200857 (2022).
63. Ren, Z., Zarepoor, M., Huang, X., Sabelhaus, A. P. & Majidi, C. Shape Memory Alloy (SMA) Actuator With Embedded Liquid Metal Curvature Sensor for Closed-Loop Control. *Front Robot AI* **8**, (2021).
64. Zadan, M. *et al.* Stretchable Thermoelectric Generators for Self-Powered Wearable Health Monitoring. *Adv Funct Mater* **34**, 2404861 (2024).
68. Engelberg, P. I. & Tesoro, G. C. Mechanical and thermal properties of epoxy resins with reversible crosslinks. *Polym Eng Sci* **30**, 303–307 (1990).
81. Sun, Q., Feng, Y., Guo, J. & Wang, C. High performance epoxy resin with ultralow coefficient of thermal expansion cured by conformation-switchable multi-functional agent. *Chemical Engineering Journal* **450**, 138295 (2022).